# Dependable Fire Detection System with Multifunctional Artificial Intelligence Framework

**DOI:** 10.3390/s19092025

**Published:** 2019-04-30

**Authors:** Jun Hong Park, Seunggi Lee, Seongjin Yun, Hanjin Kim, Won-Tae Kim

**Affiliations:** Computer Science and Engineering, Koreatech University, Cheonan-si 31253, Korea; astrada1@koreatech.ac.kr (J.H.P.); sss8412@koreatech.ac.kr (S.L.); hiysr0308@koreatech.ac.kr (S.Y.); gks359@koreatech.ac.kr (H.K.)

**Keywords:** fire detection, dependability, IoT, artificial intelligence, distributed MQTT, SDN

## Abstract

A fire detection system requires accurate and fast mechanisms to make the right decision in a fire situation. Since most commercial fire detection systems use a simple sensor, their fire recognition accuracy is deficient because of the limitations of the detection capability of the sensor. Existing proposals, which use rule-based algorithms or image-based machine learning can hardly adapt to the changes in the environment because of their static features. Since the legacy fire detection systems and network services do not guarantee data transfer latency, the required need for promptness is unmet. In this paper, we propose a new fire detection system with a multifunctional artificial intelligence framework and a data transfer delay minimization mechanism for the safety of smart cities. The framework includes a set of multiple machine learning algorithms and an adaptive fuzzy algorithm. In addition, Direct-MQTT based on SDN is introduced to solve the traffic concentration problems of the traditional MQTT. We verify the performance of the proposed system in terms of accuracy and delay time and found a fire detection accuracy of over 95%. The end-to-end delay, which comprises the transfer and decision delays, is reduced by an average of 72%.

## 1. Introduction

Dependable fire detection systems with high accuracy and promptness are essential for the safety of smart city services. Since there are so many false alarms in legacy fire detection systems, they occasionally make system operators take risky actions such as turning off low-precision fire detection systems [1]. Improving fire detection systems could prevent many accidents due to fires. During 2009–2012, excluding malicious calls 48% of all fire alarms were false alarms [2]. Of all 6,684,500 fire accidents in United States, 4,879,685 cases occurred where fire detection systems were installed [3]. Unfortunately, 20% of the fire detection systems in the USA do not correctly work [1]. 

Commercial fire detection systems generally have a simple sensor with low accuracy and are sensitive to sensor failure and malfunction which makes it hard to detect fires [4]. In addition, even if the sensor is operating normally, faults may occur in the system because of the limitations of the detection capability of the sensor. A smoke sensor may send false alarms because of the incorrect recognition of heavy dust as smoke [5]. Fire detection systems with image sensors have difficulty detecting fires in the blind spots of cameras. Therefore, it is necessary to combine heterogeneous sensors for improved fire detection performance [6]. 

The existing fire detection systems use rule-based algorithms with static parameters that make it difficult to accurately detect fire in dynamic situations [7]. Fire detection systems require suitable thresholds with specific parameters depending on the system’s installation environment. In addition, the data sensed in the same place may change with time. The dynamic threshold is essential for applying rule-based algorithms, but it is impossible for normal users to calculate these thresholds and apply them to their systems. In particular, when a fire occurs, the temperature data show a rapid increase in a short time [8,9]. Rule-based algorithms with static thresholds do not identify a fire when the sensing data do not exceed the threshold value. 

In general, artificial intelligence (AI) is suitable for identifying situations that are difficult to classify using the features extracted from the sensing data [10]. For example, AI can determine the special characteristics set or special features of various fire situations using the collected data and apply them to make decisions in actual fire events. Since machine learning (ML) is categorized as a subfield of AI, ML inherits the characteristics of AI, providing machines with the ability to make decisions by learning from data and optimizing the functional weights or parameters of algorithms instead of relying on the user to specify rules according to the situation [11]. Each ML algorithm may utilize specific types of training data depending on the purposes. For example, the Convolutional Neural Network (CNN) requires image data to recognize objects and the Recurrent Neural Network (RNN) is suitable for processing sequential data, including voice and text [12,13]. However, it is difficult for ML to adapt to dynamic changes because the values of the ML variables are fixed after training. Flexibility must be supported to compensate for the problems with ML. Since ML algorithms create different decision probabilities based on the same target, the results can incorporate the mutual-complementary factors of an ensemble method to obtain a better performance than a single algorithm [14]. Our ensemble method integrates various AI algorithms into a multifunctional AI framework that combines the advantages of each ML and has enhanced flexibility.

Combustible materials and a delayed reaction time are the main causes of fire spread, accounting for 40% and 28% of all fires, respectively [15]. Prompt fire detection effectively decreases the fire spread that follows primary causes. Fire detection systems are important for early fire detection because combustible materials create fires that spread explosively. There are two types of delayed reaction time: the time to detect a fire in a location and the time to move from a fire station to this location. If the arrival delay exceeds 5 min, property damage is doubled compared to cases with delays below 5 min [16]. Since the delay time to move to the fire site is affected by external factors, such as traffic congestion, we aim to reduce reaction delay by minimizing fire detection delays. Fire detection delays are due to two main causes: decision delays and transfer delays. Decision delays depend on the hardware performance and the complexity of the fire decision algorithms installed on the fire detection systems. Since transfer delays are subject to network congestion or specific bottleneck nodes on the forwarding paths, the latency time should be managed in the serious cases. While the transfer delays absolutely affect the promptness of the fire detection system, most fire detection systems do not consider the minimization of the transfer delays. 

Recent studies on advanced fire detection systems have been made based on IoT middlewares [17,18,19], such as MQTT, which is the common IoT middleware used by the most IoT platforms including oneM2M [20,21]. IoT middlewares like MQTT that use message queue mechanisms do not support real-time quality of service (QoS) policies because they are designed as a centralized architecture which has difficulties in supporting real-time services [22]. In spite of the problems, MQTT has been widely used in smart city services based on IoT. Therefore, studies complementing the problems are required to support the time critical services [23,24].

In this paper, we propose a dependable fire detection system using a multifunctional AI framework which interworks a novel Direct-MQTT. The main features proposed in this paper are as follows: Improved speed of fire detection by minimizing the delay of data transmission by deleting the queuing delay that occurs in the central broker when using the MQTT protocol.Combined and analyzed complex fire sensor data using multi-functional AI framework to improve fire detection accuracy.

This paper is divided as follows: in Section 2, we summarize the legacy fire detection systems, AI, and IoT middlewares. Section 3 describes the proposed fire detection system architecture and explains the multifunctional AI framework and Direct-MQTT. In Section 4, we demonstrate the performance of the proposed system in terms of the fire detection accuracy and the minimization of data transfer delay. Finally, the conclusion and future works regarding the fire detection system are provided.

## 2. Related Works

### 2.1. The Legacy Fire Detection Systems

Many studies have been conducted to combine fire detection systems with ML algorithms to analyze the images and sensor data and obtain accurate results. Shen proposed a study of flame detection using the YOLO framework [25]. YOLO is a framework that is used in various fields, including flame detection, because R-CNN-based multi-object recognition is possible. In his study, he trained models with 76% fire detection capability using 196 fire images. However, the number of fire images used for training may not be sufficient to effectively train the model. Kaabi proposed a study to detect fire smoke using a Deep Belief Network (DBN) [26]. A total of 482 data sets were used for the training and a model with 95% accuracy was produced. Since his method recognizes smoke only, it is impossible to detect fire with a certain flame recognition. Hu proposed a Neural Network of Deep Convolutional Long-Recurrent Networks (DCLRN) and conducted real-time fire detection experiments [27]. It also used a method of detecting flame movement by combining the optical flow method. He used about 10,000 images obtained from 70 fire videos in total and made a model with 93.3% accuracy. However, there are cases where faults occur from mistaking flames and lights. If he used it with other sensors, he would be able to correct those errors. Saputra proposed a fire detection system that directly implemented a fuzzy algorithm and detected fire using multiple sensors [28]. The fuzzy algorithm integrates multiple sensor data and deduces the results. However, the fuzzy algorithm has a disadvantage of using a static threshold called a fuzzy factor. S-FDS, as proposed by Jang, is a smart fire detection system that combines CNN and fuzzy logic and collects sensor data and image data using heterogeneous sensors and CCTV [29]. S-FDS preprocesses the image data using a CNN algorithm to detect fires. However, the CNN algorithm that analyzed these images could not quickly and accurately detect fires in blind spots, such as underneath a table. In particular, the CNN algorithm is difficult to use in a place where a camera cannot be used because of privacy problems, such as restrooms. In order to solve the blind spot problem of the CCTV, the fuzzy logic derives the fire recognition probability from the analyzed image data and the heterogeneous sensor data. However, S-FDS is less flexible because it is based on static, rule-based algorithms.

### 2.2. Artificial Intelligence

Recently, various algorithms have used statistical techniques for data analysis. Algorithms can have different characteristics depending on the calculation formula, and the data analysis result may vary depending on which algorithm is used. There is an ML algorithm for analyzing data. ML algorithms can be divided into shallow learning, such as Support Vector Machine (SVM), decision trees, and K-Means, as well as deep learning, which uses neural network layers such as a Deep Neural Network (DNN) and CNN [30,31,32,33,34]. The use of a deep learning model for classification is more flexible than a shallow learning model in the real environment. In general, DNN is used to determine the current value through quantified data or analyze the changing situation through continuous data. However, immediate detection is difficult because data should be collected over a period of time so that DNN may measure the changing situation. In addition, CNN is used to analyze the situation by detecting features in image data. When measuring flames in an image using video data, the image feature capture may be delayed because the flame is small if the fire has only just begun [35]. Fuzzy algorithms express a closeness to each situation that is not clearly divided using the membership function. The membership function of the fuzzy algorithm can change its range when the environment changes, but the general fuzzy algorithm ignores these changes. To solve this problem, there is an adaptive fuzzy algorithm that can update the membership function [36]. However, the adaptive fuzzy algorithm does not filter out the exception data due to sensor errors, so it does not obtain accurate results.

### 2.3. IoT Middleware

Many types of IoT middleware have been studied for their ability to connect various devices and transmit data in large-scale IoT systems. MQTT, CoAP, HTTP, XMPP are representative IoT middleware protocols [37,38,39,40]. MQTT is a standardized protocol in OASIS and can maintain the connectivity of multiple devices using a broker. Broker-based protocols may have data concentration problems and single point of failure issues because many devices connect and transmit data to the broker. MQTT limits the number of devices that are connected to a broker to 1024 to prevent queue delays because of the concentration of multiple packets. MQTT provides low-level QoS and does not consider data transfer delays, which make it difficult to use in complex network situations. DM-MQTT, as proposed by Park, considers the multiple subscribers through multicasting [41]. However, this mechanism has not solved the traffic concentration due to the rendezvous point, and it is not effective in situations with few subscribers. Santamaria [42] proposed a way to reduce the serving time of the system by using the data filter applied to the server and the MQTT protocol to process a lot of data of the ioT devices used in the e-Health system. However, this proposal cannot reduce the data transfer delay from the publishing device to the server because there is no improvement of the MQTT protocol. Santamaria proposed an anti-terrorism surveillance system [43] that can respond quickly to suspicious objects when they are found in three layers. Distributed surveillance systems are able to respond more quickly to cloud computing using edge computing technology. However, a method of applying only edge computing without considering protocol improvement cannot solve the problem of traffic overload and data concentration because of a large amount of data transmitted from multiple devices. CoAP is a standardized protocol in the IETF CoRE working group and transmits data in a RESTful method. Since CoAP is essentially designed with a peer-to-peer protocol, it has difficulty in supporting large IoT systems that require multiple edge nodes. The well-known HTTP is an application-level protocol for hypermedia information systems. HTTP transmits data in a RESTful method. Since HTTP does not consider a device with low computing power such as a sensor node, there is a problem like a big header overhead in the process of transmitting even simple sensor data. XMPP is a protocol for instant messaging developed by the Jabber open source community in 1999. XMPP is designed with scalability in mind for real-time systems and various network environments. XMPP, which is an XML message-based communication, has a distributed system such as e-mail, but has the weakness of lack of QoS and end-to-end encryption. Functional comparison of communication middleware is shown in Table 1. In addition, there are many IoT middleware such as AMQP, a lightweight peer-to-peer protocol, and Hy-LP, which fuses various protocols. In this paper, we use MQTT protocol considering the advantages of standards based protocol.

## 3. Proposed Fire Detection System Architecture

The structure of the proposed system is shown in Figure 1. The architecture consists of IoT gateways, a fire detection server, and a software-defined networking (SDN) controller. 

### 3.1. Architecture Overview

The fire detection system may be constructed as a large-scale system that collects data from multiple IoT gateways. The IoT gateways send the collected data from heterogeneous sensors, which include temperature, humidity, gas sensors, and cameras to a multifunctional AI framework that identifies the fire. The multifunctional AI framework uses multiple machine learning to analyze the heterogeneous data to calculate the fire probability for each sensor. The multiple ML algorithms include a CNN analyzing the fire image data from the camera, and a DNN processing the time series data from heterogeneous sensors. The fire probabilities for each sensor measured through MLs are combined into a single fire probability by using an adaptive fuzzy algorithm, which modifies the dynamic fuzzy factor according to the multiple ML results to infer the fire probability suitable for the environment where the system is installed. The fire detection system transmits data using the MQTT protocol, which is a broker-based middleware that can create a bottleneck problem and cause data transfer delays. The SDN controller minimizes this data transfer delay by solving the bottleneck problem with the data transmission path distribution mechanism. The SDN controller makes the direct paths between the fire detection server and the IoT gateways.

### 3.2. Multifunctional AI Framework

The multifunctional AI framework is designed to provide context adaptability, high accuracy of fire detection, and rapid decision-making by combining various AI technologies. It can accurately analyze a variety of data types, integrate the analyzed data as required by the system, flexibly adapt to changes in time and space. As mentioned above, the multifunctional AI framework consists of a CNN algorithm, a DNN algorithm, and an adaptive fuzzy algorithm. The framework has three blocks: an IoT data collection block, a context preprocessing block, and a context decision block. The overall structure of the multifunctional AI framework is shown in Figure 2.

The IoT data collection block subscribes to the MQTT session and continuously gathers heterogeneous data from the IoT gateway. In addition, it classifies and feeds the data to the context preprocessing block along with each data type. All MQTT data is stored in the database and used as input data for the sequential data analysis modules through the data preprocessing module.

The context preprocessing block has multiple ML algorithms that can be replaced with others depending on the input data types. It combines the image data analysis module with CNN and the sequential data analysis module with DNN. The multiple ML uses the ensemble method to more effectively analyze heterogeneous sensor data than a single ML would. 

The context decision block uses an adaptive fuzzy module to fuse the outputs from the context preprocessing block in order to calculate the exact probability of the fire. In addition, it can simultaneously reconfigure the fuzzy factors depending on the context. The dynamic fuzzy factors automatically change to deduce appropriate results for the target environment. We will perform experiments to compare the proposed adaptive fuzzy algorithm and other ML algorithms that may be used in the context decision block instead of the adaptive fuzzy module. 

#### 3.2.1. Image Data Analysis Module

The image recognition algorithm using CNN has proved its effectiveness and is used in various industrial fields, including fire detection services [44,45]. In this study, we also apply CNN to a multifunctional AI fire detection system (MAI-FDS) in order to enhance the performance of fire detection in terms of visual intelligence. The CNN algorithm is constructed as shown in Figure 3. It consists of five convolution layers and one fully-connected layer. The image is resized to a size of 256 × 256. Each convolution layer has a 0-padding, 5 × 5 size feature map and 2 × 2 max pooling. The depth of the feature map in the first layer is 8, which is doubled every time it passes through the hierarchy. After passing through five layers, 256 images of 8 × 8 are generated. Finally, the image is classified through a fully-connected layer of 1024 neurons.

The CNN algorithm learns two classes of fire situations and warning situations. A fire situation is determined when an actual fire is detected in the image, and a warning situation is obtained when a fire risk is recognized in the image. For example, a lighter flame does not mean an actual fire but just the probability of a fire risk. There is a score for each class, and the result is the class with the highest score. If the scores of both classes are all lower, the algorithm judges the situation as normal. Table 2 shows the training data, training frequency, and accuracy of the CNN model. The score of CNN’s fire situation class is entered as a factor of the fuzzy algorithm.

#### 3.2.2. Sequential Data Analysis Module

To compensate for the blind-spot problem of image sensors, we use a heterogeneous sensor and the DNN algorithm to analyze the sensor data [46]. The input vector of the DNN is the continuous sensor data change value, which is configured to detect a fire situation. The configuration of the DNN model is shown in Figure 4. The DNN model consists of five layers: the input layer, the output layer, and three hidden layers. The DNN model uses three types of changed input data: Final-First-Difference (FFD), which is the difference between the final value and the first value; Final-Max-Difference (FMD), which is the difference between the final value and the maximum value; and Final-Average-Difference (FAD), which is the difference between the final value and the average value. The database sends the temperature, humidity, and gas sensor data to the data-preprocessing module, which refines the data sent from the database to the input values required by the DNN model. The converted input data passes through the hidden layer, which is 100 × 100 × 100, and through the output layer to the fire, warning, and normal situations. The DNN algorithm is divided into fire, warning, and general situations. Figure 5 depicts these three situations and their different graph shapes.

These graphs show a temperature change over 1 min. The fire situation is a graphical representation of actual fire data, which shows a sudden change of more than 20 °C per minute. The normal data show almost constant graph movement and the abnormal situation shows abnormal graph movement over a short time. The training number and accuracy of the DNN algorithm are shown in Table 3.

Finally, the DNN algorithm derives the fire, warning, and normal situations. The warning situation shows only a short change in the data. The system decides if this situation is a sensor failure or a spark ignition around the sensor. Since it returns to normal data after a short period, it is not considered a fire situation, and only a warning message is sent for confirmation. The results of the fire situation and the normal situation are used in the fuzzy algorithm to calculate the final fire probability, which is identical to the CNN algorithm.

#### 3.2.3. Adaptive Fuzzy Module

Though we can identify the fire, warning, and normal situations in each CNN and DNN algorithm, we also use a fuzzy algorithm by fusing the result of sensor and image data to obtain a more accurate fire probability [47]. In particular, the fuzzy algorithm is required to obtain the precise fire probability because the sensor data are not constant. The input set is defined as temperature (T), humidity (H), gas (G), image (V) and fire (F). Each input set is defined as shown in Figure 6. 

The fuzzy algorithm uses a membership function that corresponds to the fuzzy set to obtain the membership grade of the fuzzy rule. Each member function can be expressed as a graph according to its argument value. Figure 7 shows an example of a fuzzy set membership function.

The membership function of the general fuzzy algorithm is divided into probability based on the data on the horizontal axis of the graph. The fuzzy algorithm that finds the grade of membership based on a fixed threshold is difficult to apply to a variety of environments [48]. If the humidity is low or a certain amount of gas is generated, there is a high possibility that a continuous false detection may occur. In particular, since the temperature continuously changes, it is possible to decide that the situation is a fire situation using the fixed threshold value even if a fire is not present. Therefore, the fuzzy algorithm must change the threshold value according to the situation. The fuzzy factor is the data that distinguishes between the low and high values of the grade of membership. The adaptive fuzzy algorithm is constructed by changing the fuzzy factor.

In general, the fuzzy factor of the adaptive fuzzy algorithm is updated to adapt to situation change on the basis of the suitable formula [49]. Although the adaptive fuzzy algorithm needs to update the fuzzy factor depending on the situation, it is necessary to ignore the fuzzy factor change when the fire data or warning data as shown in Figure 5 is input. We propose an improved adaptive fuzzy algorithm that can more flexibly determine fuzzy factor changes than existing adaptive fuzzy algorithms. In the proposed MAI-FDS, the result of the context preprocessing block is used to decide whether to update the fuzzy factor. The update process of the fuzzy factor is as follows. All data transmitted from the IoT gateway is stored in the database. When data analysis is completed through the DNN algorithm in the context preprocessing block, the result is sent to the fuzzy factor change function. If the DNN algorithm decides that the situation is normal, the fuzzy factor change function requests the average data from the database, which calculates the average value of the data for the last five minutes and replies. The fuzzy factor change function sets the average value as the fuzzy factor and derives the final fire probability through the fuzzy algorithm. The DNN algorithm does not change the fuzzy factor if it decides that the result is a fire or a warning situation. In the case of a fire situation, the data show a sudden change over five minutes and increases the average value. Therefore, if the fuzzy factor is changed by this average value, the final fire probability is inaccurate. In addition, when the sensor obtains data such as temperature or humidity, unpredicted exception data that may affect the accuracy of the algorithm may be received. Such anomalous data may bad affect the update in the fuzzy factor. If the DNN algorithm detects the exception data by analyzing the obtained data, it is not applied to the update of the fuzzy factor. The fuzzy membership function that is redefined by the fuzzy factor change function is shown in Figure 8.

When the fuzzy factor changes, the membership function of each fuzzy set changes according to the change in the target environment. Therefore, accurate fire probabilities can be calculated through dynamic criteria. For example, let us suppose that the fuzzy factor of the temperature membership function is 40. When the average temperature of the environment is 10 °C, the membership function of the existing fuzzy algorithm does not change. However, the adaptive fuzzy algorithm membership function is changed to 10 °C by changing the fuzzy factor. In this case, when a fire occurs and the temperature change suddenly increases to 30 °C, the existing fuzzy algorithm has a low fire probability because the fuzzy factor is 40 °C, but the probability of the adaptive fuzzy algorithm is high. Thus, the adaptive fuzzy algorithm enables precise and speedy fire measurement in a variety of environments.

### 3.3. Direct-MQTT

The fire detection delay time can be divided into the decision delay and the transfer delay. The decision delay is the computation time for the calculation of the fire probability depending on the complexity of the fire detection algorithm and the computing power of the fire detection server. The transfer delay includes processing delay, transmission delay, propagation delay and queuing delay on SDN switches. In addition, the queuing delay on a MQTT broker is an important consideration for time critical IoT services. The standard MQTT protocol is based on a broker [50]. All publishers and subscribers are connected to the broker to transmit data. This mechanism does not require the Discovery process because the publisher and the subscriber do not have to check each other’s location, which solves the problem of traffic overload caused by discovery. However, the data transmitted to the broker can be concentrated, which may cause traffic congestion and consequently, data transfer delay. An example of the existing MQTT protocol traffic flow using an SDN is shown in Figure 9.

Figure 9 shows the path of the data centered on the central broker. The broker queuing delay is added to the data transfer delays because of the data concentration and the delay time (*T_total_delay_*) is calculated as in (1): (1)$$T_{total\_delay}=T_{decision}+T_{transfer}+T_{broker\_queuing}$$

Since the value of $T_{broker\_queuing}$ may be measured by tens of seconds, reducing the broker queuing delays should be essential in order to speed up the fire detection system. We propose Direct-MQTT as a novel MQTT mechanism to minimize the broker queuing delays. The mechanism solves the traffic concentrated on the broker by functionally integrating the SDN controller and the extended MQTT broker. The structure of the integrated SDN controller is shown in Figure 10.

The SDN controller adds an extended MQTT block to distribute the flow of data during the routing process. The standard MQTT protocols transmit data using topics. Publishers and subscribers are connected through the broker and transfer messages. The publisher sends the topic that contains data to the broker, and the subscriber receives the topic from the broker. The broker stores the subscriber’s IP address and information on their topic subscription. The SDN controller collects the information on the subscribers stored in the broker. The SDN controller determines the flow of all MQTT packets based on their information to prevent transfer delay. The transmission process of each packet is shown in Figure 11.

Publishers and subscribers are connected through the broker and transfer messages. The subscriber sends a subscribe message to the broker to notify it of the subscription topic. The broker sends the subscription topic and IP address of the subscriber to the SDN controller, which creates the subscriber information table in the topic management module of the extended MQTT block. The publisher sends the collected data to the broker and the SDN switch sends the first incoming data to the SDN controller for routing. The MQTT Packet Receiver Module in the extended MQTT block checks the MQTT packet and sends it to the topic-matching module, which matches the topic of the packet information with the topic on the subscriber information table. After the topic is matched, the emergency data recognition module identifies whether it is emergency data. The dest_ip switch module changes destination address of the packet and the SDN controller sets its path. Finally, the packet setting module sends the data of the set route and the destination address to the SDN switch, which determines the flow of the MQTT packet through the configuration data transmitted from the SDN controller. Since the changed destination address is the subscriber’s IP address, the data sent from the publisher is delivered to the subscriber instead of the broker. As a result, traffic distribution minimizes data transfer delays by eliminating queuing delays in the broker.

An example of the traffic flow of the proposed MQTT protocol is shown in Figure 12. Compared with Figure 9, the flow of MQTT data in Figure 12 is more distributed. In Figure 9, only the optimal path to the central broker is set, although the SDN is used. In addition, the broker and subscribers are set to a single path with a one-to-one connection. However, in Figure 12, an SDN controller with an extended MQTT block distributes the broker’s centralized data path, creating a path directly to the subscriber. Minimizes data transfer delays by eliminating queuing delays in the broker by distributing traffic to all data from the publisher, not to the broker but directly to the subscriber.

## 4. Analysis

In this section, we perform fire detection and data transfer delay tests to verify the MAI-FDS. In order to verify the accuracy and speed of the algorithm, we compare MAI-FDS and EFDS fire probability graphs. The data transfer delay tests measure the latency of standard MQTT, CoAP, and Direct-MQTT in a complex network environment.

### 4.1. Experimental Environment

Actual fire data are required to verify the accuracy and promptness of MAI-FDS. Experiments with real fires are limited because of costs and accident risks. In addition, several environmental variables make it impossible to generate identical fire effects. To solve this problem, we used a virtual environment that repeatedly reproduced the characteristics of an actual fire. The Fire Dynamics Simulator (FDS) [51], which is an open-source fire simulation developed by the U.S. National Institute of Standards and Technology (NIST), has proven its reliability in previous studies. The simulated data of the FDS, in which the model is finely structured, showed errors within 10% of the actual fire data [52,53,54]. We used the FDS to generate fires in a virtual environment to obtain temperature, humidity, and gas sensor data. Table 4 shows the types of fuel and fuel sources in the virtual environment generated by the FDS. The required data for the image analysis algorithms used a fire video generated in a similar space as the environment configured in the FDS. The network testbed is based on the Mininet SDN emulator [55]. As the SDN controller, we use the Ryu controller [56] to manage the entire network flow. Figure 13 shows the network topology, which simulated the scenarios in Figure 9 and Figure 12 in this experiment. 

Figure 13 shows the network topology with 24 SDN switches. The total number of hosts was 18. On the left, there are the 10 hosts that acted as the publisher, which serves as a gateway for collecting sensor data. As the MQTT broker, we used a Mosquitto broker [50] in the center of the network topology. On the right, there are the 7 connected hosts that acted as the subscriber. All switches were connected to the SDN controller to control the network flow. Based on the topology in Figure 13, the number of published samples and subscriber hosts changed and the test was performed. In the experimental environment, all paths were specified as the minimum path using Dijkstra’s algorithm [57].

### 4.2. Fire Detection Algotrithm

In order to verify the speed of the fire decision algorithms used in MAI-FDS, three algorithms were compared: the temperature threshold algorithm, the CNN-based algorithm, and the early fire detection system (EFDS) [28]. The fire decision algorithm uses temperature, humidity, gas, and image data as decision data. The temperature threshold algorithm is based on a simple decision algorithm by temperature data and determines a fire if the temperature exceeds 50 °C. The CNN-based algorithm uses image data to determine fire. The EFDS configured with a fuzzy algorithm uses both temperature, humidity, and gas sensor data, but it does not use image data. The experiment data are shown in Figure 14. 

The experiment was carried out over six minutes. The DNN and fuzzy algorithms set an adaptive threshold based on five minutes of data. MAI-FDS calculates the current fire probability by combining newly collected data and accumulated 5-min data. Figure 14 shows the change in the experimental data. In the video data, the fire was reduced to 131 s, the temperature to 132 s, the gas concentration to 138 s, and the humidity to 136 s. Using this data, the fire probability graph per second is shown in Figure 15.

In this paper, we did not consider a situation where a sensor fails or the system is turned off and cannot detect a fire. We first verify the speed of fire detection with the assumption that the sensor and system will operate normally. The fire detection time of the temperature threshold algorithm is the slowest due to limitations of the detection capability of the sensor because it only analyzes temperature data. The CNN algorithm is fast at initial detection, but because of image quality limit, it takes a long time for the probability of fire to rise to a certain level. The EFDS using both temperature, humidity, and gas sensor data is similar to MAI-FDS. However, MAI-FDS recognizes fire by 6 s faster than the EFDS algorithm when the fire detection threshold is set to 40%. By increasing the fire threshold to 90%, MAI-FDS is 16 s faster than the CNN.

The difference in fire detection time between the other algorithms and MAI-FDS occurs in the processing of sensor data and adaptive fuzzy algorithms. Unlike algorithms that use static rules, MAI-FDS uses z deep-learning algorithm and can train changes in temperature and change the threshold value of the fire decision. When calculating fire probability using temperature data, the temperature threshold algorithm does not show a high probability until it exceeds 50 °C. MAI-FDS shows a high probability of fire if the temperature sharply increases even before it reaches 50 °C. In Figure 14, the temperature shows a rapid change of 10 °C for 30 s between 130 and 160 s. Since the temperature was less than 50 °C at 160 s, the temperature threshold algorithm showed a low fire probability for the sensor data. MAI-FDS detects sudden changes in data and reconfigures the temperature threshold value of the decision algorithm. MAI-FDS outputs high fire probabilities due to updated thresholds even at low temperatures. The EFDS that detect low humidity data recognize fire faster than the temperature threshold algorithm that uses only temperature data. The EFDS is also slower than MAI-FDS because of the limitations of rule-based algorithms. When the fire probability threshold for fire alarm is set to 90%, the fire alarm occurrence time of each algorithm is as follows: MAI-FDS is 12 s; EFDS is 14 s; the CNN algorithm is 72 s; and the temperature threshold algorithm is 62 s.

In order to verify the fire recognition flexibility of the adaptive fuzzy algorithm, we compare fire probabilities of various algorithms using the restaurant data during cooking activities. The temperature, humidity and gas data of a restaurant steeply fluctuate as the cooking process. When the cooking begins, all sensor data will rise. Sometimes the humidity and gas data show opposite results depending on the item being cooked. The identification of fire in a kitchen can be made by the changing data in a short time. Gas data can increase drastically under cooking, but temperature and humidity data are plotted with a gentle slope. If a fire actually occurs, all the sensing data including temperature, humidity and gas will show corrupt changes in a specific period. The adaptive fuzzy algorithm recognizes this difference, changes the threshold value, and can decide the fire situation according to the changed threshold even when the similar patterns of temperature, humidity, and gas data are received. The input data for the experiments are temperature, humidity and gas data generated in the restaurant for 24 h. The image data are excluded from the input data because it is not affected by the adaptive fuzzy algorithm. The data assume a situation in which cooking takes place from 10:00 to 15:00, 16:00 to 22:00, and a fire occurs at 11:00. Changes in data during the cooking process show a gradual change due to kitchen ventilation. Data in a fire situation is a time when the ventilation system does not work and shows a rapid change due to an explosion of fire. The probability of fire recognition over time is shown in Figure 16. The temperature threshold algorithm often recognize fire because more than 50 °C of temperature data are detected during cooking. Most of the cooking process is highly humid, but if a lot of gas is generated under specific circumstances, the EFDS maybe consider situation as a fire due to high temperature and high gas. Since the adaptive fuzzy algorithm of MAI-FDS analyzes the data for five minutes, the data in the cooking process with gentle slope is not identified as a fire. The fuzzy factor is adjusted with the variation of the data, and the probability of a fire does not change even if the temperature exceeds the threshold value, in this case 50 °C.

### 4.3. Data Tranmission Experiments

The experiments of the transfer delay time for HTTP, CoAP, the standard MQTT and Direct MQTT are made in terms of scalability, that is to say, the number of publishers and subscribers. The experimental environment is shown in Figure 13. The number of switches was 24, and the location of the broker was set to the center of the topology. The network congestion for the propagation delay, queuing delay, and transfer delay was set to 5 ms per path. To test a situation in which the quantity of sensor data increased, we increased the number of publishers from the host, which acts as a gateway. In this experiment, each host increased the number of publishers by 25, starting from one. Finally, the subscriber collected up to 1000 samples and calculated the average delays. 

Figure 17 shows the transfer delay test results for the fire data. The standard MQTT showed increasing latency when it sent more than 500 samples. The standard MQTT uses a broker to transmit data, resulting in a broker queue delay. The CoAP protocol also shows a similar transfer delay because it transmits data through a central server. HTTP has higher latency than standard MQTT and CoAP due to header overhead. Since Direct-MQTT eliminates queue delays in the broker by distributing the data transmission path, it can transmit data 30% faster than the standard MQTT. However, the Direct-MQTT also showed that when many publishers send data, the delay increases with the network congestion. 

The experiments showed that an increase in the number of publishers is based on a single subscriber. Therefore, if the broker is near the subscriber, the standard MQTT obtains a similar result to the Direct-MQTT. However, in a network configuration where there are multiple subscribers, the location of the broker cannot be set near a specific subscriber. Since the IoT system transmits data between multiple publishers and subscribers, it is necessary to check the transfer delay in multi-subscriber situations. In this experiment, we configured the network topology as the default setting. It consisted of 24 switches, a centrally located broker, and 25 publishers from 10 hosts transmitting data. The congestion level of the network was set to 5 ms. We measured the transfer delay by increasing the number of subscribers from one to seven. We assumed that all publishers transmitted the data to all subscribers.

Figure 18 shows the data transmission delay with an increasing number of subscribers. For multiple subscribers, the standard MQTT publisher sent the data to the broker without further action. Additional operations on the standard MQTTs were performed on the broker. For a single subscriber, the broker deletes the data from the queue immediately after it sends the data. However, with multiple subscribers, the broker increases the processing delays because the process of sending data to all subscribers is completed and the data are removed from the queue. When there were seven subscribers, an average delay of 8 s was measured with a maximum delay of 52 s. The HTTP protocol and the CoAP protocol have additional latency in the process of sending data from a single server to multiple clients. Distributed servers solve latency issues, but many servers have management difficulties and cost issues. Direct-MQTT had a shorter total delay time than the standard MQTT because there was no broker queuing delay. 

By comparing the Direct-MQTT delay times in Figure 17 and Figure 18, we may note that the total delay increases when the number of subscribers increases in the same network topology. In Figure 17, the measured total delay was 0.3 s, but in Figure 18, it was 3 s. In the Direct-MQTT, each publisher used unicast to directly send data to multiple subscribers. The Direct-MQTT has the disadvantage of increasing the number of unicast routes due to the increase in number of publishers, subscribers, and final delay because of the network congestion in the path.

As described in Equation (1), the end-to-end delay is the sum of the decision delay and the data transfer delay, or the time between fire occurrence and detection. In order to compare the end-to-end delay, we simulated a fire probability measurement experiment where data was collected from 750 heterogeneous sensors. The network topology of the experiment is shown in Figure 13 and the data in the experiment is shown in Figure 14. MAI-FDS transmits the data using Direct-MQTT and the others transmit the data using the standard MQTT. The experimental results of all systems are compared in Figure 19. The calculated decision delay of MAI-FDS was 5 and 13 s end-to-end, and the decision delay of CNN algorithm was 12 s and 72 s end-to-end, as shown in Figure 15. The data transfer delay of Direct-MQTT is nearly 0 s, and the data transfer delay of the standard MQTT only exceeds 2 s when 750 sensor samples are sent, as shown in Figure 17. Thus, the end-to-end delays of MAI-FDS are approximately 5 s and 13 s, and the end-to-end delays of fuzzy algorithm are approximately 14 s and 74 s. 

The calculated end-to-end delays of each system are similar to the measurement delays, as shown in Figure 19. As a result, MAI-FDS is approximately 72 % faster than fuzzy algorithm.

## 5. Conclusions

Fire detection systems should consider two important engineering factors: fire detection probability and transfer delay. Since the existing fire detection systems use rule-based algorithms or simple AI algorithms, there are low detection accuracy problems caused by different environmental conditions, in addition to little usage of heterogeneous sensors. In addition, the legacy systems have difficulty satisfying promptness requirements because they do not consider the transfer delays of fire events caused by traffic congestion on a specific node or a server. In this paper, we proposed the usage of MAI-FDS to solve these problems. MAI-FDS adopts a multifunctional AI framework with an IoT data collection block, a context preprocessing block, and a context decision block. Each functional block in the framework has high flexibility depending on its applications. MAI-FDS showed much higher fire recognition capability than the legacy FDSs based on rules, CNN, and fuzzy methods. In addition, the adaptive fuzzy algorithm reduces the false alarms by calculating the amount of data change. In experiments using restaurant data, there were cases where static algorithms were decided as a fire at the time of cooking. MAI-FDS, which uses an adaptive fuzzy algorithm, did not deduced these as a fire at the time of cooking because the data showed no rapid change. Incorporating Direct-MQTT in MAI-FDS improved the speed of the fire detection system through its path distribution mechanism. The SDN controller supported by Direct-MQTT created direct paths between each publisher and the corresponding subscribers. This direct path mechanism effectively reduced the transfer delays by eliminating the queuing delays in the broker. As a result, MAI-FDS achieved an average fire detection accuracy of above 95% and reduced the end-to-end transfer delay by 67% compared with that of the existing fire detection systems. We believe that the research results of MAI-FDS will be an important system of future smart cities and will help prevent the spread of fires through accurate fire detection. Accurate fire detection and reduced latency of critical alarm events improve the reliability of the disaster mitigation system. In a future smart city where all infrastructures can be intelligent, a reliable disaster mitigation system will effectively support rapid firefighting measures such as optimizing the route to the fire scene. However, MAI-FDS adapts in all environments, making it difficult to accurately determine the fire. Adaptive fuzzy algorithms and multifunctional AI frameworks have been designed for environmentally adaptive fire detection systems, but the ability to detect fires in extreme environments is still weak. If a melting furnace is used in a factory, the temperature is very high and there is a continuing presence of flames, which can increase the false positive rate in MAI-FDS. Therefore, in future research, it is necessary to develop a system that can modify the fire detection algorithm and develop a multifunctional AI framework to adaptively detect the fire in the real environment.

## Figures and Tables

**Figure 1 sensors-19-02025-f001:**
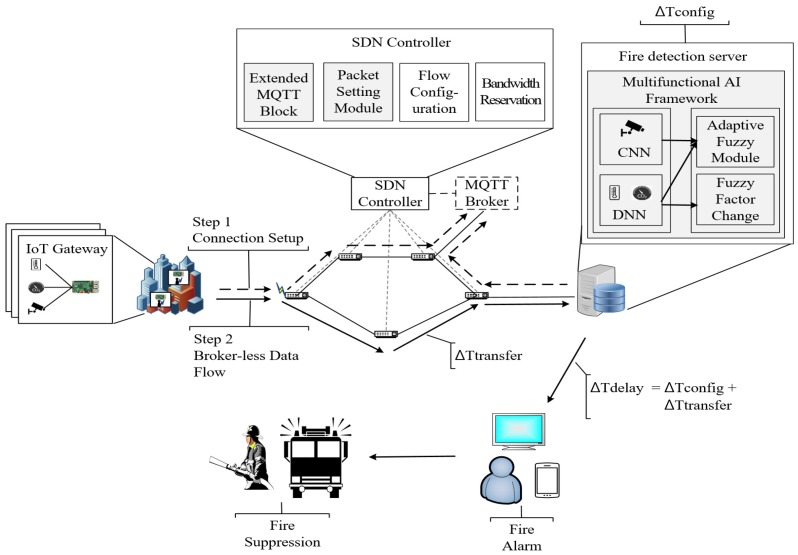
Dependable fire detection system with a multifunctional AI framework.

**Figure 2 sensors-19-02025-f002:**
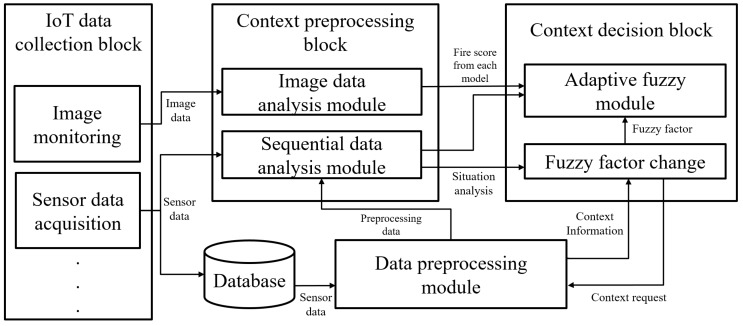
The proposed multifunctional AI framework.

**Figure 3 sensors-19-02025-f003:**
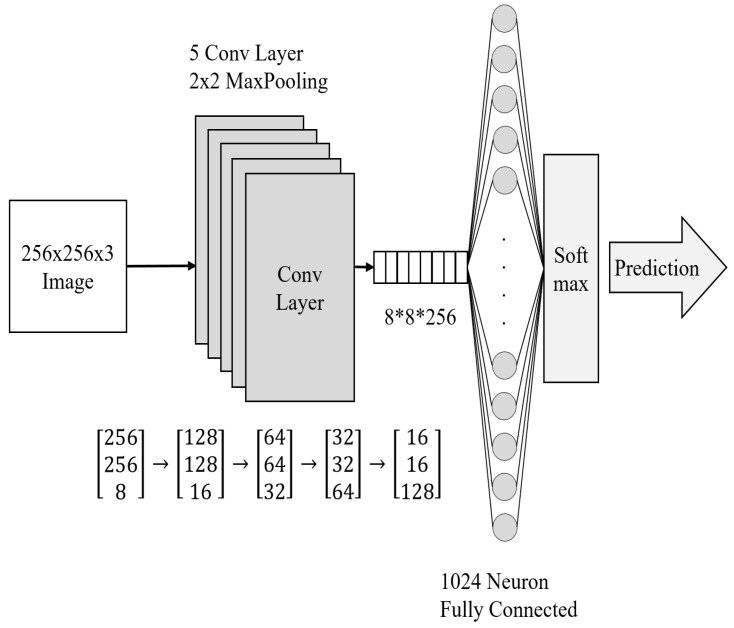
The proposed Convolutional Neural Network architecture to analyze the image data.

**Figure 4 sensors-19-02025-f004:**
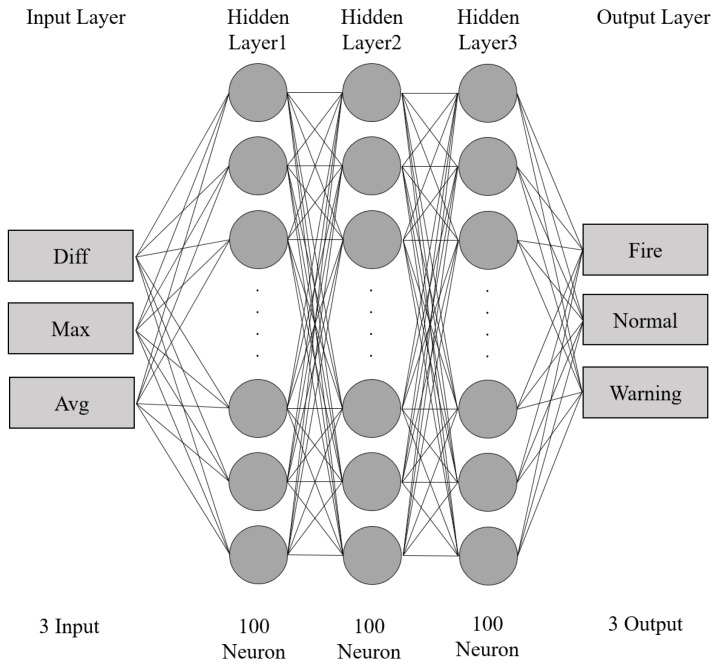
The proposed Deep Neural Network architecture for sequential architecture.

**Figure 5 sensors-19-02025-f005:**
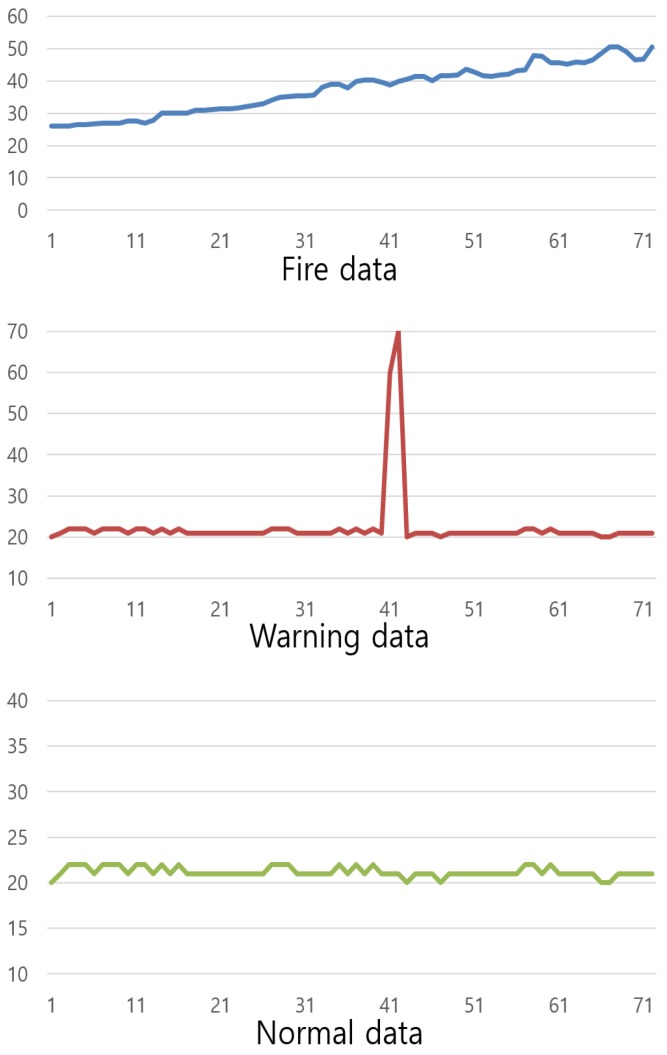
Learning data patterns classified into three classes.

**Figure 6 sensors-19-02025-f006:**
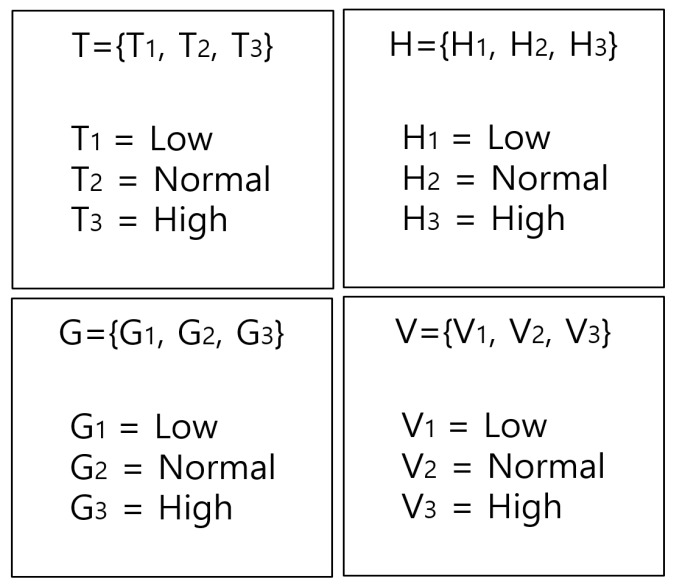
Fuzzy set definition of the adaptive fuzzy algorithm.

**Figure 7 sensors-19-02025-f007:**
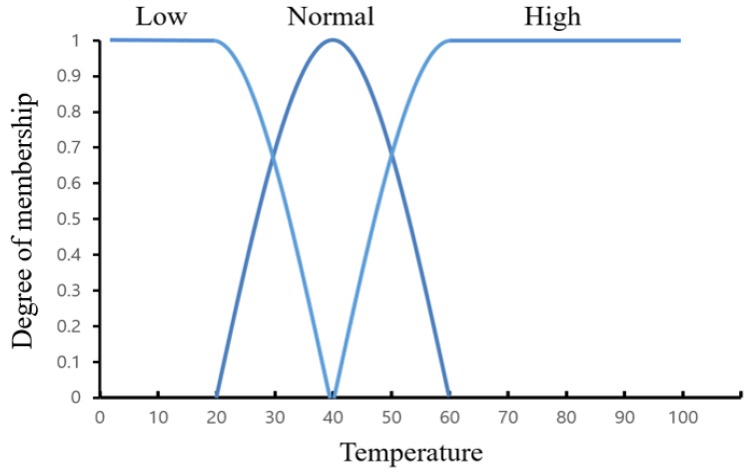
Example of a temperature-dependent function in general fuzzy algorithms.

**Figure 8 sensors-19-02025-f008:**
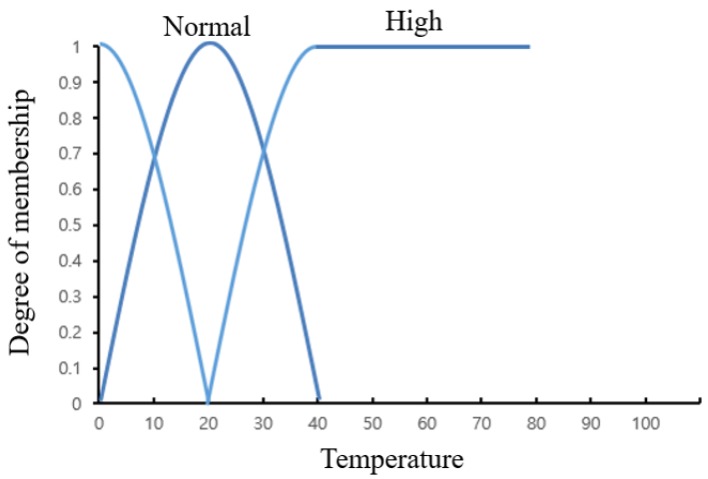
Example of a modified fuzzy membership function.

**Figure 9 sensors-19-02025-f009:**
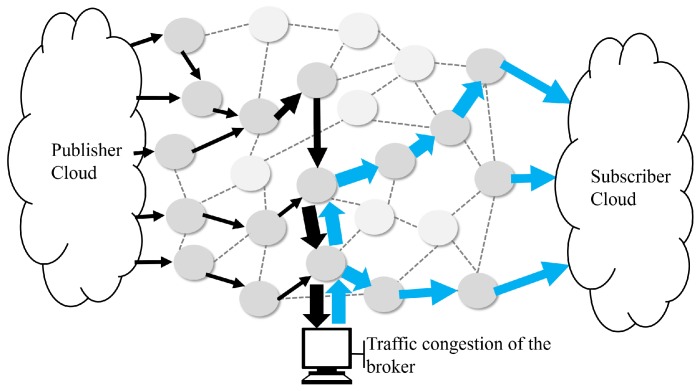
Example of the existing MQTT data transfer path in a mesh topology.

**Figure 10 sensors-19-02025-f010:**
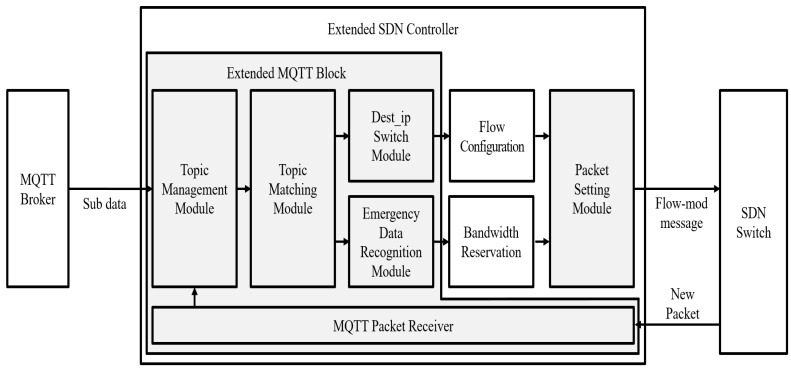
The proposed SDN controller using the extended MQTT Block.

**Figure 11 sensors-19-02025-f011:**
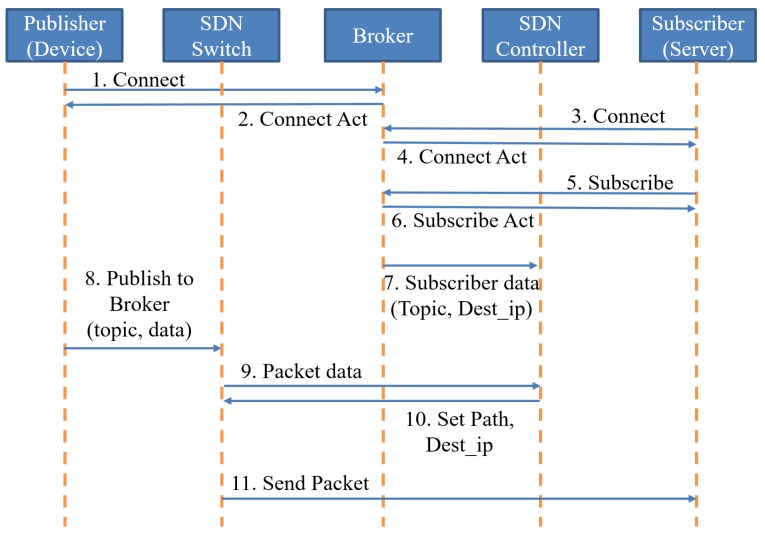
Data transfer sequence diagram of Direct-MQTT.

**Figure 12 sensors-19-02025-f012:**
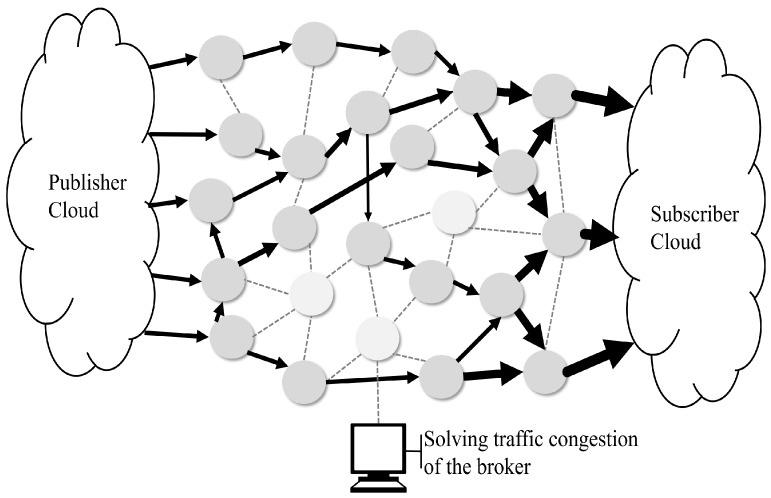
Example of the Direct-MQTT data transfer path in a mesh topology.

**Figure 13 sensors-19-02025-f013:**
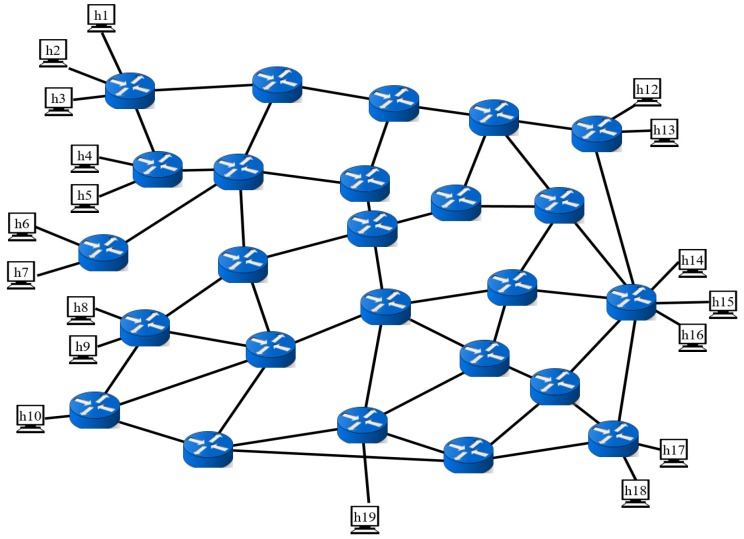
Network topology used in the experiments.

**Figure 14 sensors-19-02025-f014:**
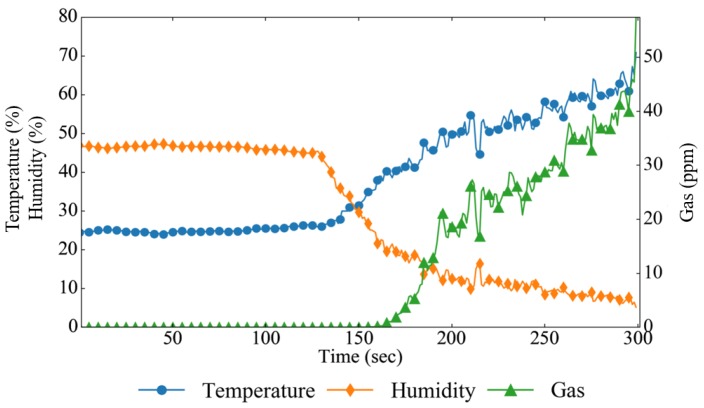
Changes in temperature, humidity, and gas data in the experiment.

**Figure 15 sensors-19-02025-f015:**
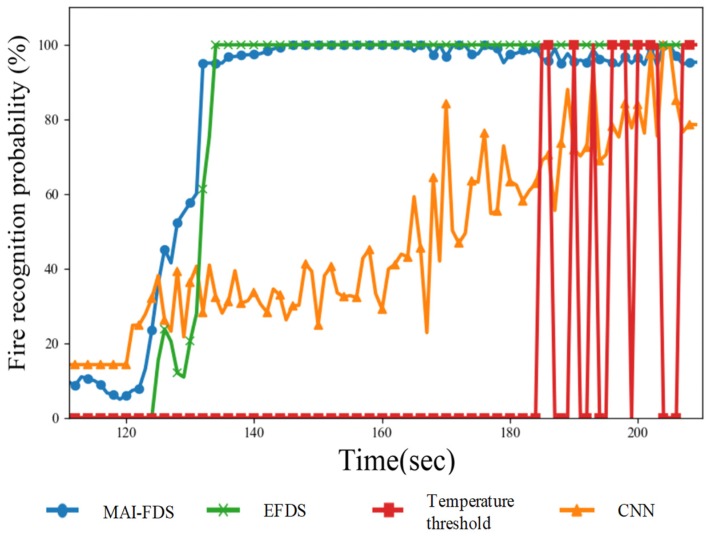
Measured fire probability variation of MAI-FDS, The CNN algorithm, the EFDS and the temperature threshold algorithm.

**Figure 16 sensors-19-02025-f016:**
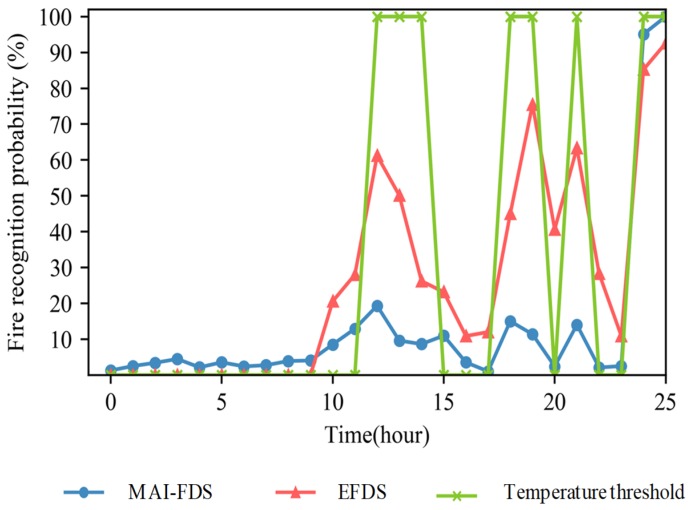
Fire recognition probabilities of MAI-FDS with adaptive fuzzy, the EFDS and the temperature threshold algorithm.

**Figure 17 sensors-19-02025-f017:**
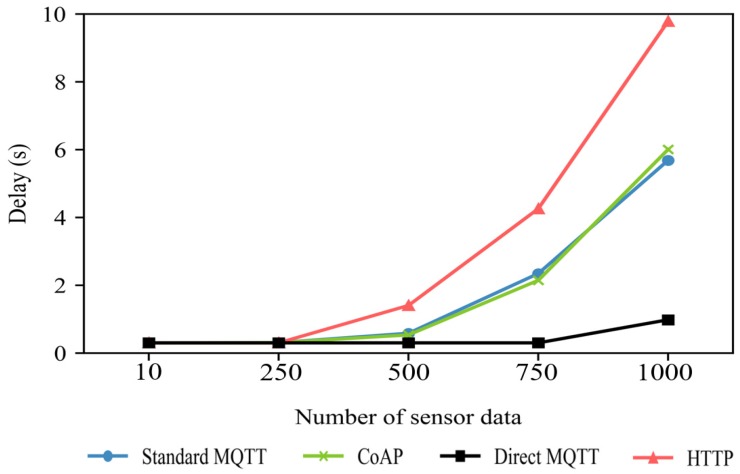
Changes in data transfer delay depending on the number of fire samples.

**Figure 18 sensors-19-02025-f018:**
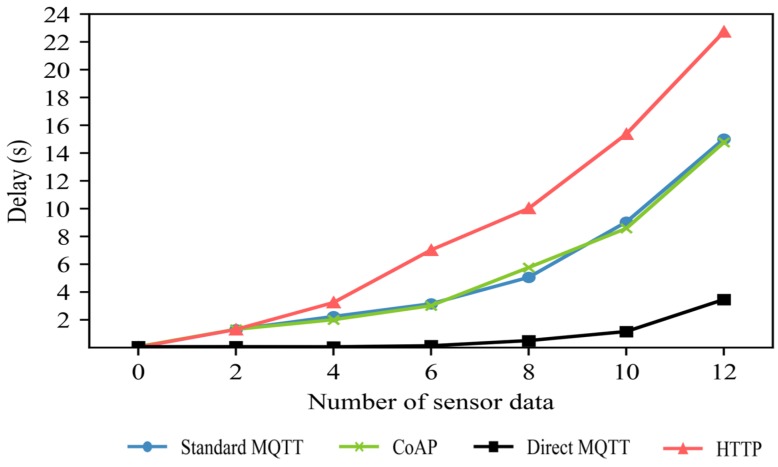
Changes in data transfer delay depending on the number of subscribers.

**Figure 19 sensors-19-02025-f019:**
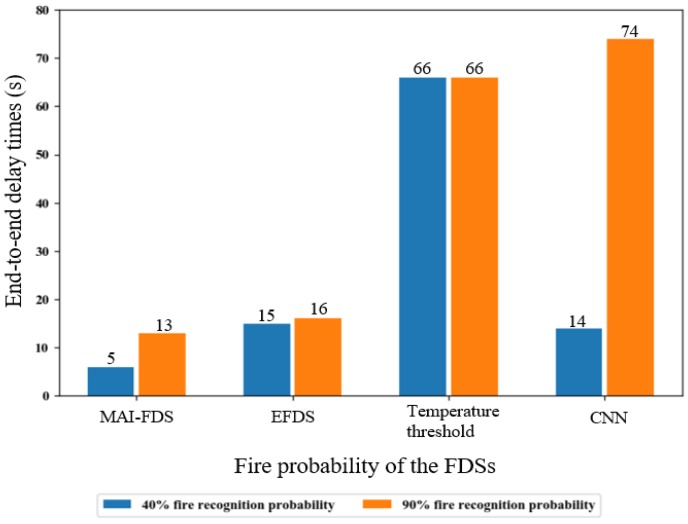
Total end-to-end delay time according to the fire probability of each.

**Table 1 sensors-19-02025-t001:** Functional Comparisons of Communication Middleware.

	MQTT	CoAP	HTTP	XMPP
Basis protocol	TCP	UDP	TCP	TCP
Data transmission	Pub/Sub	RESTful	RESTful	Pub/Sub
Quality of Service	Three policies	Not supported	-	Not supported
Support scale	Large scale	Middle scale	Large scale	Large scale
Communication architecture	Centralized	Centralized	Centralized	Distributed

**Table 2 sensors-19-02025-t002:** CNN Model training set and accuracy.

	Training Data	Testing Data
Fire	8500	300
Matches	1119	300
Accuracy	90.3%	91%
Training Steps	14000	-

**Table 3 sensors-19-02025-t003:** DNN Model training set and accuracy.

	Training Data	Testing Data
Normal data	10,000	1000
Fire data	10,000	1000
Warning data	3000	300
Accuracy	99%	99%
Training Steps	69,000	-

**Table 4 sensors-19-02025-t004:** Fire source and fuel source in the simulation.

**Fuel Source**	**Fuel Name**	**Chemical Formula**
	N-HRPTANE	Heptane, C7H16
**Fire Source**	**Object Name**	**Elements of Object**
	Gas range frame	PLASTIC (75%) STEEL (25%)
	Futon/Pillow	FABRIC
	Sofa	FOAM (80%) FABRIC (20%)

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
