# Peer review of "Dependable Fire Detection System with Multifunctional Artificial Intelligence Framework"

_sensors, 2019, doi:10.3390/s19092025_

Round 1
Reviewer 1 Report
The article presents a complex proposal, based on diverse technologies, to provide an early warning system in case of fire.
In general, the paper is well written, presented, and its structure is mature. But in my opinion, it presents a great shortcoming: there is no serious comparison, in a real scenario or simulated on the basis of real histories, where it is demonstrated that this proposal is superior to other existing ones. The architecture is complex, the IA system intricate, and all this can remain in a set of buzzwords if it does not demonstrate the real gain of this proposal compared to others, which should have been studied in depth in a more complete section dedicated to the state of the art.
Author Response
Overall comment: The article presents a complex proposal, based on diverse technologies, to provide an early warning system in case of fire.
Response: Thank you for reviewing the posting changes. Through revising, we had a good experience about writing papers after reviewing the details of our paper environment to be applied. Most of the reviews you gave modified to reflect in our paper.
Comment 1-1: In general, the paper is well written, presented, and its structure is mature. But in my opinion, it presents a great shortcoming: there is no serious comparison, in a real scenario or simulated on the basis of real histories, where it is demonstrated that this proposal is superior to other existing ones. The architecture is complex, the IA system intricate, and all this can remain in a set of buzzwords if it does not demonstrate the real gain of this proposal compared to others, which should have been studied in depth in a more complete section dedicated to the state of the art.
Response: Thank you for a good review comment. As explained in Section 4.1, we determined that it is very dangerous to conduct an experiment by generating an actual fire. So we thought about using a simulator. We would not have used a simulator if there was a lot of difference between the actual fire sensor data and the simulator fire data. However, NIST's FDS is a fire simulator that is used in various papers and projects, and is highly accurate simulator for fire reproduction. In our laboratory, we compared the actual fire data with the FDS data and found that they showed more than 90% accuracy. This information can be found in S-FDS, which was created as a reference in the paper. In this paper, we performed experiments using the data analyzed in the S-FDS. We believed that the results we tested in this paper will be similar to those verified in real scenarios.

Reviewer 2 Report
Main Comments:
This paper presents a fire-detection system that utilizes a broker-less MQTT variation for IoT messaging and multifunctional artificial intelligence (AI) for reasoning. Novelties in both domains are described. The authors also perform a comparative analysis with relevant AI detection algorithms in a simulated environment.
However, several issues are not clear in the text and must be clarified by the authors.
- The paper has several similarities with other papers that have been previously published by the same authors, like the article that is mentioned below and was published in the same journal in 2018:
DM-MQTT: An Efficient MQTT Based on SDN Multicast for Massive IoT Communications, Sensors 2018, 18(9), 3071; doi:10.3390/s18093071
The authors must clearly state the differences from their previous work, change similar analysis, discussions, figures, etc.
- The notion of ‘broker-less MQTT’ seems confusing. You should use a different description.
- Highlight the main novelties of your proposal as a bullet-list at the end of the introductory section.
- In the introduction, you pay much attention in the false positives. For fire detection systems, safety is the man concern. Thus, false positives are important, but not critical. The most significant factor is the elimination of false negatives. That’s way there is a high volume of false alarms in the literature. You should better present this issue in the text. Also, in the evaluation sections, you should mention the false negative ratios for the evaluated settings.
- In the ‘2.3. IoT middleware’ subsection, you mention MQTT, CoAP, and HTTP as IoT middlewares. CoAP is actually a lightweight version of HTTP that was designed for IoT. In general, HTTP is nor referenced as IoT middleware. Try to cite other messaging protocols that are more suitable for the IoT ecosystem, e.g. XMPP, AMQP, Hy-LP, etc.
- In Table 1, why you mention that the communication architecture for CoAP and HTTP is centralized?
- In the ML description, you mention that you try to detect fire as a temperature change over 1 minute (i.e. 20oC increase per minute). In another paragraph you state that you monitor the sudden changes over 5 minutes. Aren’t these time slots quite high for early fire detection?
- In figure 12, it is not clear in the figure why the data flows change. You should better illustrate the SDN modules and depict the differences from figure 9. Also, no traffic seems reaching the broker.
- In the analysis section, you mention that you include in the comparative study the Emergent-MQTT, but you do not.
- In figure 13, the node ids are not depicted well.
- The main document ends a little bit sharply. You should add a section of future work, mentioning the integration of you proposal with disaster mitigation systems that manage the system after the fire detection and assist in the emergency management, e.g. AmbiSPDM, DOMAPS, DEFACTO, ALADDIN, etc.
- Check the references and comply with the journal’s format in all cases.
- The use of English is decent. Nevertheless, minor corrections are needed in the text.
Other:
- Define the terms CNN, RNN, QoS, SVM, DNN, MAI-FDS, SDN, S-FDS, EFDS
- Provide references for ‘Since there are so … due to fire’
- In the sentences ‘During 2009-2012 … were installed [2]’, specify the place.
- Change ‘by combining optical flow method ‘ into ‘by combining the optical flow method’
- Change ‘using multiple sensor‘ into ‘using multiple sensors’
- Change ‘K-Means, and deep learning‘ into ‘K-Means, as well as deep learning’
- Change ‘has difficulty supporting‘ into ‘has difficulty in supporting’
- Change ‘Well-known HTTP‘ into ‘The well-known HTTP’
- Change ‘3. Proposed Fire Detection System Arcjitecture‘ into ‘3. The Proposed Fire Detection System Architecture’
- Change ‘be construct with a‘ into ‘be constructed as a’
- Change ‘and cameras, to a‘ into ‘and cameras to a’
- Provide a reference for ‘MAI-FDS’
- Change ‘the performance of fire detection capability‘ into ‘the performance of fire detection’
- Change ‘change value is configured to detect‘ into ‘change value, which is configured to detect’
- Change ‘input data :‘ into ‘input data:’
- Change ‘FFD (Final-First-Difference)‘ into ‘Final-First-Difference (FFD)’
- Change ‘FMD (Final-Max-Difference)‘ into ‘Final-Max-Difference (FMD)’
- Change ‘FAD (Final-Average-Difference)‘ into ‘Final-Average-Difference (FAD)’
- Change ‘. figure 5‘ into ‘. Figure 5’
- Change ’20 oC‘ into ‘20oC’ (remove the space). Apply the change in all related cases.
- Change ‘decides this situation is‘ into ‘decides if this situation is’
- Change ‘. figure 7‘ into ‘. Figure 7’
- Change ‘in Fig. 5 is input‘ into ‘in figure 5’
- Change ‘if it decides the result‘ into ‘if it decides that the result’
- Change ‘3.2 Direct MQTT‘ into ‘3.3 Direct MQTT’
- Change ‘delay time(T)‘ into ‘delay time (T)’
- Change the notation (T) and use different symbols for ‘delay time’ and ‘temperature’
- Change ‘to minimize broker‘ into ‘to minimize the broker’
- Change ‘1:1‘ into ‘one-to-one’
- Provide the reference for ‘S-FDS’ (e.g. [29])
- Provide the reference for ‘Emergent-MQTT’ (e.g. [40])
- Provide references for the ‘Mininet SDN emulator’ and the ‘Ryu controller’
- Change ‘. figure 13‘ into ‘. Figure 13’
- Change ‘On the left are‘ into ‘On the left, there are’
- Provide the reference for ‘Mosquitto broker’ (e.g. [47])
- Change ‘On the right are‘ into ‘On the fight, there are’
- Provide the reference for ‘Dijkstra algorithm’
- Change ‘the speed of fire detection‘ into ‘the speed of the fire detection’
- Change ‘various algorithms uses the‘ into ‘various algorithms using the’
- Change ‘the threshold value in this case‘ into ‘the threshold value, in this case’
- Change ‘4.2 Data transmission experiments‘ into ‘4.3 Data transmission experiments’
- Change ‘starting with one‘ into ‘starting from one’
- Change ‘faster than standard MQTT‘ into ‘faster than the standard MQTT’
- Change ‘increase in number of‘ into ‘increase in the number of’
- Change ‘help prevent‘ into ‘help preventing’
Author Response
sensors-447607: “Dependable Fire Detection System with Multifunctional Artificial Intelligence Framework”
Response to reviewers’ comments:
Recommendation: Pending major revisions
Reviewer 2
Overall comment: This paper presents a fire-detection system that utilizes a broker-less MQTT variation for IoT messaging and multifunctional artificial intelligence (AI) for reasoning. Novelties in both domains are described. The authors also perform a comparative analysis with relevant AI detection algorithms in a simulated environment.
Response: We are grateful for spending your precious time on our review of the paper. Our paper has received good help and has a chance to rethink your review comment. Most of the reviews you gave modified to reflect in our paper. The revised context was marked in green.
Comment 2-1: The paper has several similarities with other papers that have been previously published by the same authors, like the article that is mentioned below and was published in the same journal in 2018:
DM-MQTT: An Efficient MQTT Based on SDN Multicast for Massive IoT Communications, Sensors 2018, 18(9), 3071; doi:10.3390/s18093071
The authors must clearly state the differences from their previous work, change similar analysis, discussions, figures, etc.
Response: We are grateful for your comments that disadvantage of our paper. DM-MQTT aims to reduce data transfer delay in large-scale IoT communications. DM-MQTT applies the multicast method to standard MQTT considering the scenario where a large number of subscribers receive data. However, this paper applies the direct routing method to the standard MQTT in consideration of environment with few subscribers such as fire detection system and other monitoring systems. The goal of this paper is to minimize the transfer delay that occurs when lots of publishers transmit data to a small number of subscribers. Multicast and direct routing are very different. DM-MQTT with CBT(Core Based Tree) selects rendezvous points that minimize transfer delays between publishers and all (many) subscribers. The data sent by the publisher goes to the rendezvous point along the same path. The data arriving at the rendezvous point will be sent to the individual routes to each subscriber. D-MQTT considers only minimal transfer delays between individual publisher and subscriber for routing. We think that multicast mechanism is not suitable for this paper because there are fewer subscribers (or one), and D-MQTT sets up a one-to-one path between subscribers and publishers, regardless of the number of publishers. The D-MQTT with all the paths distributed is different from the multicast method that requires rendezvous points.
Comment 2-2: The notion of ‘broker-less MQTT’ seems confusing. You should use a different description.
Response: We receive to grateful your review. We have once again considered the meaning of the broker-less MQTT and agree that it is not appropriate. For the above reasons, the broker-less MQTT was deleted and only the Direct-MQTT is used.
Comment 2-3: Highlight the main novelties of your proposal as a bullet-list at the end of the introductory section.
Response: We would like to thank you for your review of the paper. We added the main function of this paper to the final paragraph of the introduction.
Comment 2-4: In the introduction, you pay much attention in the false positives. For fire detection systems, safety is the man concern. Thus, false positives are important, but not critical. The most significant factor is the elimination of false negatives. That’s way there is a high volume of false alarms in the literature. You should better present this issue in the text. Also, in the evaluation sections, you should mention the false negative ratios for the evaluated settings.
Response: We are grateful for the recommendation of our paper. We understood false negative as a situation where a fire occurred but the sensor could not detect it. Existing fire detection systems will eventually detect fire as time elapses when a fire occurs. Only when the fire detection system is shut down or the sensor is broken, the fire can not be detected to the end. However, if the fire can`t be detected quickly, the fire detection system becomes meaningless. So we thought that false negative can be solved by the ability to detect fire quickly. There are many ways to detect fire quickly. First, there is a way to use a sensor with good performance. You can use expensive sensors such as flame detection sensors or infrared cameras. Or you can use temperature and gas sensors with wide sensing range and good recognition rate. However, we have written a paper to improve the speed of fire detection with algorithms using data collected from ordinary sensors rather than improving sensors.
Comment 2-5: In the ‘2.3. IoT middleware’ subsection, you mention MQTT, CoAP, and HTTP as IoT middlewares. CoAP is actually a lightweight version of HTTP that was designed for IoT. In general, HTTP is nor referenced as IoT middleware. Try to cite other messaging protocols that are more suitable for the IoT ecosystem, e.g. XMPP, AMQP, Hy-LP, etc.
Response: We appreciate your paper comments on our deficient paper. We added XMPP in section 2.3, which describes the IoT protocols.
Comment 2-6: In Table 1, why you mention that the communication architecture for CoAP and HTTP is centralized?
Response: Thank you for your revising comments. CoAP is basically a structure in which one server and one node participate in 1: 1 communication. We, like the brokers of MQTT, expressed the CoAP is a centralized architecture because of the presence of server.
Comment 2-7: In the ML description, you mention that you try to detect fire as a temperature change over 1 minute (i.e. 20oC increase per minute). In another paragraph you state that you monitor the sudden changes over 5 minutes. Aren’t these time slots quite high for early fire detection?
Response: Thank you for reviewing the posting changes. The DNN algorithm needs a baseline value to detect sudden changes in data. In this paper, as described in Section 3.2.2, the DNN algorithm uses FFD, FMD, and FAD data to determine the fire. These data are calculated from the data of the last 5 minutes based on the most recently collected sensor data. As shown in Figure 5, the fire data shows a rapid change for one minute. At this time, the FFD that calculates the difference between the first data and the last data of 5 minutes of data shows a big difference. However, FAD, which calculates the 5-minute average, does not differ greatly. Judging from these changes, the DNN algorithm determines the fire. Therefore, the previous 5 minutes of data is required at the time of collection. However, this 5-minute data means 5 minutes of data before the fire occurs to determine the current fire and does not mean 5 minutes of data after the fire occurred. The goal of this paper is to quickly check the changing sensor data when a fire occurs after accumulating 5 minutes of data before a fire occurs.
Comment 2-8: In figure 12, it is not clear in the figure why the data flows change. You should better illustrate the SDN modules and depict the differences from figure 9. Also, no traffic seems reaching the broker.
Response: Thank you for a good review comment. Figure 9 and Figure 12 represent data flow only and the process of changing is shown in Figure 10 and Figure 11. The basic MQTT protocol shows the data flow in Figure 9, and the results from the process described in Figure 10 and Figure 11 are shown in Figure 12. In Figure 9 and Figure 12, adding an SDN controller made the picture too complicated, and the change process was not explained well in the drawing. Therefore, we added Figure 10 and Figure 11 and selected only the results shown in Figure 12. In addition, we have added additional information in Figure 12 to help you understand Figure 12.
Comment 2-9: In the analysis section, you mention that you include in the comparative study the Emergent-MQTT, but you do not.
Response: We receive to grateful your review. We compared the Emergent-MQTT and the S-FDS in our experiments when we started writing the paper. However, we found that Emergent-MQTT and S-FDS were not suitable for comparison. S-FDS was written in the same laboratory and Emergent-MQTT was also not suitable for comparison because the author is the same. Therefore, we changed to another comparison algorithm and resumed the experiment. Because the comparison algorithm had changed, the content had to change, but there were some mistakes left in the beginning of section 4. So we changed this part to fit the paper.
Comment 2-10: In figure 13, the node ids are not depicted well.
Response: Thank you for your revising comments. We made the mistake of thinking that location is more important than node ids. Therefore, we changed the size of the picture so that the ID of the node can be seen clearly.
Comment 2-11: The main document ends a little bit sharply. You should add a section of future work, mentioning the integration of you proposal with disaster mitigation systems that manage the system after the fire detection and assist in the emergency management, e.g. AmbiSPDM, DOMAPS, DEFACTO, ALADDIN, etc.
Response: Thank you for reviewing the posting changes. We are going to change the fire detection algorithm in future research. We did not consider a system for fire suppression after a fire, but we thought it necessary to reconstruct the fire algorithm and study how to detect the fire adaptively in the real world. This was added at the end of the conclusions.
Comment 2-12: Check the references and comply with the journal’s format in all cases.
Response: We would like to thank you for your review of the paper. We re-confirmed all the references and modified them to match the journal`s format.
Comment 2-13: The use of English is decent. Nevertheless, minor corrections are needed in the text.
Response: Thank you for a good review comment. I have applied all the comments you wrote in the ‘other’ comments. I also confirmed the whole paper again.

Round 2
Reviewer 1 Report
In my opinion the validation could be improved.
Author Response
sensors-447607: “Dependable Fire Detection System with Multifunctional Artificial Intelligence Framework”
Response to reviewers’ comments:
Recommendation: Pending major revisions
Reviewer 1
Overall comment: In my opinion the validation could be improved.
Response: Thank you for reviewing the posting changes. The comparison between the FDS simulation data and actual fire data was conducted in a number of papers. When the model of the FDS is elaborately constructed, the error of the simulation data is verified to be less than 10% by the references [50-52] which we added for the fact check. In this paper, since we have a local legal issue for actual fire test as well, we constructed a sophisticated model to make the FDS environments and the FDS configuration information is defined in Table 4. Difficulties in actual fire testing are described in Section 4.1 and references to reliability of FDS simulation data are added in Section 4.2. Modified situations are shown in red.

Reviewer 2 Report
The authors addressed some of my comments, but not all of them.
- I insist that there are still several similarities with their previous paper and the related content must be altered (e.g. figures, analysis, discussions, etc.).
- Include in the text the main point of your answers for the comments 2-1, 2-4, and 2-7.
- For the IoT protocols, include a small paragraph surveying the state-of-the-art solutions, as there are several different solutions like AMQP, Hy-LP, etc., and justify why you concentrate in the specific protocols that you examine in your study.
- For the discussion regarding the timing (the minutes which are required in the analysis phase) clarify in the text how much time it takes to your system to detect a fire and raise an alarm, once the fire has started.
- At the end of the document, add one or two paragraphs outlining the operation of disaster mitigation systems that can manage the system after the fire has been detected.
Author Response
sensors-447607: “Dependable Fire Detection System with Multifunctional Artificial Intelligence Framework”
Response to reviewers’ comments:
Recommendation: Pending major revisions
Reviewer 2
Overall comment: The authors addressed some of my comments, but not all of them.
Response: We are grateful for spending your precious time on our review of the paper. Our paper has received good help and has a chance to rethink your review comment. Most of the reviews you gave modified to reflect in our paper. The revised context was marked in green.
Comment 2-1: I insist that there are still several similarities with their previous paper and the related content must be altered (e.g. figures, analysis, discussions, etc.).
Response: We are grateful for your comments that disadvantage of our paper.
There are some difference between DM-MQTT and D-MQTT in terms of functional and structural aspects. DM-MQTT is based on a multicast mechanism because it is efficient to transmit the same sensing data to multicast groups. DM-MQTT has applied a shared tree mechanism for effective data transfer and group management. However, DM-MQTT did not solves the traffic concentration problem on a single point, that is to say, on the rendezvous point of DM-MQTT, caused by the shared tree multicasting. Furthermore, the multicast mechanism is not efficient for a small number of subscribers, and data transmission delays may be large because of the single point traffic concentration. In this paper, D-MQTT solves the problem by eliminating the single point of traffic concentration problem. D-MQTT minimizes data transfer delays by adopting a fully distributed message transport mechanism through direct SDN paths between the publishers and the subscribers instead of multicast trees of DM-MQTT.
In addition, there are functional difference between D-MQTT and DM-MQTT. DM-MQTT constructs a hierarchical MQTT broker structure with a master broker and multiple slave brokers to manage multicast groups. The SDN controller of DM-MQTT communicates with the master broker that collects data from each slave broker to analyze location information, and topic information of all publishers, subscribers, and slave brokers. After DM-MQTT considers each slave broker, it forms a multicast group of publishers/subscribers and specifies the optimal rendezvous point for them. Unlike DM-MQTT, D-MQTT does not require the hierarchical MQTT broker structure and the location of each broker is not important. D-MQTT creates the shortest path between a publisher and a subscriber with the same MQTT topic in order to minimize transfer delay.
Finally, we recognized the similarities with the previous paper in Figure 10 and Figure 13. We did our best to make changes as you recommended.
Comment 2-2: Include in the text the main point of your answers for the comments 2-1, 2-4, and 2-7.
Response: We receive to grateful your review. Thank you for your review. I have added answers for 2-1, 2-4 and 2-7 comments on pages 4 (“DM-MQTT, as proposed by Park, considers the multiple subscribers through multicast [41]. However, this mechanism has not solved the traffic concentration due to the rendezvous point, and it is not effective in situation with few subscribers.”), at page 14 (“The DNN and fuzzy algorithms set an adaptive threshold based on five minutes of data. MAI-FDS calculates the current fire probability by combining newly collected data and accumulated 5-minute data.”) and at page 15 (“When the fire probability threshold for fire alarm is set to 90%, the fire alarm occurrence time of each algorithm is as follows: MAI-FDS is 12 seconds; EFDS is 14 seconds; the CNN algorithm is 72 seconds; and the temperature threshold algorithm is 62 seconds.”), respectively, depending on your opinion.
Comment 2-3: For the IoT protocols, include a small paragraph surveying the state-of-the-art solutions, as there are several different solutions like AMQP, Hy-LP, etc., and justify why you concentrate in the specific protocols that you examine in your study.
Response: We would like to thank you for your review of the paper. MQTT has shown advantages in many-to-many communications and was thought to be suitable for IoT systems as a lightweight protocol. In addition, the MQTT protocol is based on the ISO standard and is specified as oneM2M standard protocol. These features have become important reasons for choosing the MQTT protocol in this paper. We added the description of AMQP and Hy-LP and the reason we chose MQTT to the related works: as follows at page 4 (“In addition, there are many IoT middleware such as AMQP, a lightweight peer-to-peer protocol, and Hy-LP, which fuses various protocols. In this paper, we use MQTT protocol considering the advantages of standards based protocol.”)
Comment 2-4: For the discussion regarding the timing (the minutes which are required in the analysis phase) clarify in the text how much time it takes to your system to detect a fire and raise an alarm, once the fire has started.
Response: We are grateful for the recommendation of our paper. We have added an algorithm-specific fire alarm on page 15 as follows: “When the fire probability threshold for fire alarm is set to 90%, the fire alarm occurrence time of each algorithm is as follows: MAI-FDS is 12 seconds; EFDS is 14 seconds; the CNN algorithm is 72 seconds; and the temperature threshold algorithm is 62 seconds.”.
Comment 2-5: At the end of the document, add one or two paragraphs outlining the operation of disaster mitigation systems that can manage the system after the fire has been detected.
Response: We appreciate your paper comments on our deficient paper. We have added information on future disaster mitigation systems on page 19 as follows: “We believe that the research results of MAI-FDS will be an important system of future smart cities and will help prevent the spread of fire through accurate fire detection. Accurate fire detection and reduced latency of critical alarm events improve the reliability of the disaster mitigation system. In a future smart city where all infrastructures can be intelligent, a reliable disaster mitigation system will effectively support rapid firefighting measures such as optimizing the route to the fire scene.”
